

# Antioxidant nutrition in Atlantic salmon (*Salmo salar*) parr and post-smolt, fed diets with high inclusion of plant ingredients and graded levels of micronutrients and selected amino acids

Kristin Hamre[1,2], Nini H. Sissener[1], Erik-Jan Lock[1], Pål A. Olsvik[1], Marit Espe[1], Bente E. Torstensen[1], Joana Silva[3], Johan Johansen[4], Rune Waagbø[1,2] and Gro-Ingunn Hemre[1]

[1] National Institute of Nutrition and Seafood Research (NIFES), Bergen, Norway
[2] Department of Biology, University of Bergen, Bergen, Norway
[3] Biomar AS, Trondheim, Norway
[4] GIFAS, Inndyr, Norway

Corresponding author
Kristin Hamre, kha@nifes.no

## ABSTRACT

The shift from marine to plant-based ingredients in fish feeds affects the dietary concentrations and bioavailability of micronutrients, amino acids and lipids and consequently warrants a re-evaluation of dietary nutrient recommendations. In the present study, an Atlantic salmon diet high in plant ingredients was supplemented with graded levels of nutrient premix (NP), containing selected amino acids, taurine, cholesterol, vitamins and minerals. This article presents the results on the antioxidant nutrients vitamin C, E and selenium (Se), and effects on tissue redox status. The feed ingredients appeared to contain sufficient levels of vitamin E and Se to cover the requirements to prevent clinical deficiency symptoms. The body levels of α-tocopherol (TOH) in parr and that of Se in parr and post-smolt showed a linear relationship with dietary concentration, while α-TOH in post-smolt seemed to be saturable with a breakpoint near 140 mg kg$^{-1}$. Ascorbic acid (Asc) concentration in the basal feed was below the expected minimum requirement, but the experimental period was probably too short for the fish to develop visible deficiency symptoms. Asc was saturable in both parr and post-smolt whole body at dietary concentrations of 190 and 63–89 mg kg$^{-1}$, respectively. Maximum whole body Asc concentration was approximately 40 mg kg$^{-1}$ in parr and 14 mg kg$^{-1}$ in post-smolt. Retention ranged from 41 to 10% in parr and from −206 to 12% in post-smolt with increasing NP supplementation. This indicates that the post-smolts had an extraordinarily high consumption of Asc. Analyses of glutathione (GSH) and glutathione disulphide (GSSG) concentrations and the calculated GSH based redox potentials in liver and muscle tissue, indicated only minor effects of diets on redox regulation. However, the post-smolt were more oxidized than the parr. This was supported by the high consumption of Asc and high expression of gpx1 and gpx3 in liver. Based on the present trials, the recommendations for supplementation of vitamin C and E in diets for Atlantic salmon are similar to current practices, e.g. 150 mg kg$^{-1}$ of α-TOH and 190 mg kg$^{-1}$ Asc which was the saturating concentration in parr. Higher concentrations than what would prevent clinical deficiency symptoms are necessary

to protect fish against incidents of oxidative stress and to improve immune and stress responses. There were no indications that the Se requirement exceeded the current recommendation of 0.3 mg kg$^{-1}$.

## INTRODUCTION

The shift from marine to plant-based ingredients in fish feeds leads to changes in dietary concentrations and bioavailability of several nutrients. So far, research on plant-based diets for Atlantic salmon has mainly focused on balancing the amino acid profile and reducing the amounts of antinutrients (*Espe et al., 2006*; *Espe et al., 2007*; *Krogdahl, Hemre & Mommsen, 2005*), while micronutrients have received far less attention. In the present study, two feeding experiments were conducted on salmon, one with parr in freshwater and one with post-smolt in seawater, using a basic diet containing 10% fishmeal and 3.5% fish oil of the total recipe, in addition to plant-based protein and lipid sources. A nutrient premix (NP), containing potentially critical micronutrients, amino acids, taurine and cholesterol, was added at graded levels to this diet and health indicators and biomarkers were used to estimate optimal supplementation of the different nutrients. *Hemre et al. (2016)* presents data on general fish performance, retention of macronutrients, B-vitamins and amino acid metabolism. Data on selected minerals and vitamin A, D and K are presented by E.-J. Lock, 2016, unpublished data. The present publication is related to antioxidant nutrients and redox regulation.

Vitamin E is the general term for a group of lipid soluble antioxidants, tocopherols and tocotrienols, where α-tocopherol (α-TOH) has the highest biological activity (United States Pharmacopeia 1993). α-TOH is selectively retained in the body of mammals (*Kayden & Traber, 1993*; *Rigotti, 2007*) and Atlantic salmon (*Hamre, 2011*; *Hamre, Berge & Lie, 1998*), possibly due to a liver TOH binding protein, with high affinity for this isomer (*Kayden & Traber, 1993*; *Yoshida et al., 1992*). Vitamin E breaks the chain of lipid peroxidation by reducing fatty acid peroxide radicals, preventing oxidation of new fatty acids (*Buettner, 1993*). It takes approximately three months for Atlantic salmon parr to adjust their body concentration of α-TOH to the dietary concentration and there is a linear relation between dietary and whole body concentrations when dietary α-TOH ranges between 0 and 300 mg kg$^{-1}$. The slope is quite reproducible between experiments, at approximately 0.25 between fish concentration on wet weight and feed concentration on dry weight (*Hamre et al., 1997*; *Hamre & Lie, 1995*). Vitamin E is added as α-tocopheryl acetate (α-TOAc) in fish feed, thereby protected against oxidation. The requirement is highly variable, in rainbow trout (*Oncorhynchus mykiss*) dietary requirements between 5 and 100 mg kg$^{-1}$ have been reported, dependent on dietary composition and experimental conditions. Factors that affect the vitamin E requirement are dietary polyunsaturated fatty acid (PUFA), vitamin C and Se concentrations and oxidized feed (*Hamre, 2011*). Furthermore, the estimated requirement

varies with the response variable used. Hemoglobin (Hb) concentration is a sensitive indicator of overall health and growth performance while indicators of immune function respond at higher dietary levels of vitamin E and will often give very high requirement assessments (*Hamre, 2011*). In the present study, γ-TOH is used as a representative of the non-α-TOHs, since it is often present in plant oils at high concentrations.

In classical nutrition, vitamin C (ascorbic acid (Asc)) has a well defined role as a cofactor for the enzymes catalyzing the hydroxylation of proline and lysine, necessary for formation of collagen and bone matrix (*Barnes & Kodicek, 1972*; *Gould et al., 1960*; *Terova et al., 1998*). The function of Asc is to keep the iron present at the enzymes' active site in the reduced state (*Meister, 1994*). More generally, Asc is a water-soluble antioxidant, it scavenges free radicals and probably participates in recycling oxidized vitamin E (*Hamre et al., 1997*; *Tappel, 1962*). The resulting Asc radical or dehydroascorbic acid can in turn be reduced by glutathione (GSH) and ultimately by NADPH produced in energy metabolism (*Mårtensson & Meister, 1991*). It was not until stable and fully bioavailable forms of vitamin C were developed that meaningful requirement studies in fish could be designed (*Woodward, 1994*). *Sandnes, Torrissen & Waagbø (1992)* performed one of the first of such studies in Atlantic salmon using Ca ascorbate-monophosphate (AP). They found a minimum dietary requirement, based on growth, mortality and skin and backbone hydroxyproline concentration, of 10–20 mg kg$^{-1}$. Severe deficiency symptoms appeared first after 18 weeks of feeding the non-supplemented diet. Concentration of Asc in the liver was linearly related to dietary AP up to 160 mg kg$^{-1}$ Asc equivalents. The appearance of vitamin C deficiency symptoms depends on the dietary concentration of vitamin E and is probably affected by other variations in the experimental conditions as well (*Dabrowski et al., 2004*; *Gabaudan & Verlhac, 2001*; *Sandnes, Torrissen & Waagbø, 1992*). It is common for several fish species that dietary Asc required for maximum body or tissue storage surpasses the requirement levels for growth, survival and hydroxyproline concentrations (*NRC, 2011*). Furthermore, immune response indicators are stimulated by Asc levels far above conventional requirements, which also protect against stress (*Trichet, 2010*; *Waagbø, 1994*; *Waagbø, 2006*).

Selenium (Se) is an essential trace element inserted in selenocysteine (Sec), situated at the active site of Se dependent proteins, termed selenoproteins (*Brigelius-Flohé, 1999*). The selenoproteins can be grouped in stress-related and housekeeping proteins, the first group responds readily to dietary Se and includes GSH peroxidase (GPX) 1 and 3 (*Penglase et al., 2014*). In mammals, the dietary Se level at which GPX1 activity and gene expression level off, is used as an indicator of the Se requirement (*Sunde et al., 2009*; *Weiss et al., 1996*; *Weiss et al., 1997*). GPX1 activity in rainbow trout plasma followed a similar trajectory to that in mammalian tissues and leveled off at dietary Se levels (given as Na$_2$SeO$_3$) between 0.15 and 0.38 mg kg$^{-1}$ (*Hilton, Hodson & Slinger, 1980*). However, in zebrafish, maximum growth correlated with minimum whole body GPX1 activity and mRNA expression at 0.3 mg kg$^{-1}$ dietary Se. Dietary Se above this level reduced growth and increased GPX1 activity and expression (*Penglase et al., 2014*). This may indicate that the relation between GPX1 and Se requirement is different in fish and mammals.

There was a linear relationship between dietary Se in the range 0.1–1.0 mg kg$^{-1}$ and whole body Se concentrations in the zebrafish study (*Penglase et al., 2014*).

The cellular redox environment affects the cell's fate because gene expression, protein function and molecular pathways are often redox sensitive (*Go & Jones, 2013*; *Jones & Sies, 2015*; *Kirlin et al., 1999*). The key mechanism is that protein cysteine residues switch between oxidized and reduced states corresponding to active and inactive states of the involved proteins (redox switches). An example is that a more oxidized cellular environment induces a redox switch that releases the nuclear factor-erythroid 2-related factor 2 (NFE2L2) transcription factor from a complex with another protein, Kelch Like ECH Associated Protein 1 (KEAP1). Once released, NFE2L2 induces the transcription of at least 50 mammalian genes; many of which code for antioxidants, thiol oxidoreductases and GSH synthesis/recycling genes; that are involved in maintaining the redox balance and/or are involved in redox signaling (reviewed by *Ma, 2013*). Cellular redox homeostasis is thought to be maintained by redox couples that act as electron buffers due to their ability to readily cycle between oxidized and reduced forms (*Jones & Sies, 2015*). The major redox couples are reduced/oxidized glutathione (2GSH/GSSG), cysteine (2Cys/CySS) and thioredoxin (Trx(SH)$_2$/TrxSS) (*Huseby, Sundkvist & Svineng, 2009*) and the related redox potential (E) is proportional to the ratio between the (reduced)$^2$ and oxidized forms of the redox couples, according to the Nernst Equation. GSH/GSSG is present in cells in high concentrations. It may therefore be the most important cellular redox couple and is used as an indicator of tissue redox state in the present study. The GSH based redox potential seems to be strictly regulated in Atlantic salmon (*Hamre et al., 2010*). A hypothesis of the present study is therefore that GSH/GSSG concentrations and $E_{GSH}$ are stable in healthy fish and can be used as indicators of fish welfare during the growth phase in salmon. On the other hand, the 2GSH/GSSG redox couple and many genes coding for proteins that maintain the cellular redox system are dynamically regulated during fish embryonic and larval development (*Hamre et al., 2014*; *Penglase et al., 2015*; *Skjærven et al., 2013*; *Timme-Laragy et al., 2013*).

Here we present data on antioxidant nutrients and redox regulation of two feeding trials (freshwater and seawater) with graded NP levels. Tissue concentrations and retention of vitamin C, E and Se, traditionally viewed as the main antioxidant nutrients, as well as effects of a graded NP on tissue GSH/GSSG concentrations and the related redox potentials, were analyzed. We have also measured the expression of some central genes for regulation of redox homeostasis in the liver; the transcription factor *nfe2l2*, *cuznsod*, *mnsod*, *cat*, *gpx1* and *gpx3* which metabolize superoxide anions and H$_2$O$_2$, *gclc* which translates into the rate limiting protein in GSH synthesis, and *g6pd* and *gr* involved in keeping GSH in the reduced state.

## MATERIALS AND METHODS

### Experimental diets

The seven experimental diets were produced at Biomar Technology Centre (Denmark), as described in *Hemre et al. (2016)*. All diets contained the same basal mixture of ingredients (10.4% marine protein ingredients and 3.5% fish oil of the whole receipt, the

**Table 1 Feed formulation Trial 1, and with slight difference given for Trial 2 in parenthesis (Trial 2).** Nutrient premix, methionine, taurine and cholesterol were added to the diets in graded amounts, and balanced by reducing the content of field peas in the diets, all other ingredients were used in equal amounts in all diets. Numbers are in g kg$^{-1}$.

| Composition | 0NP | 25NP | 50NP | 100NP | 150NP | 200NP | 400NP |
|---|---|---|---|---|---|---|---|
| Fish meal SA 68 superprime | 80 | 80 | 80 | 80 | 80 | 80 | 80 |
| Krill meal | 24.2 | 24.2 | 24.2 | 24.2 | 24.2 | 24.2 | 24.2 |
| Soy Prot. Conc. 60% | 180 | 180 | 180 | 180 | 180 | 180 | 180 |
| Corn gluten 60 | 40 | 40 | 40 | 40 | 40 | 40 | 40 |
| Pea protein 75 | 124 (130) | 124 (130) | 124 (130) | 124 (130) | 124 (130) | 124 (130) | 124 (130) |
| Wheat gluten | 180 (150) | 180 (150) | 180 (150) | 180 (150) | 180 (150) | 180 (150) | 180 (150) |
| Wheat | 61 (60) | 61 (60) | 61 (60) | 61 (60) | 61 (60) | 61 (60) | 61 (60) |
| Field peas | 100 | 98 | 95 | 90 | 85 | 80 | 60 |
| Fish oil, capelin | 35 (44) | 35 (44) | 35 (44) | 35 (44) | 35 (44) | 35 (44) | 35 (44) |
| Rapeseed oil | 79 (88) | 79 (88) | 79 (88) | 79 (88) | 79 (88) | 79 (88) | 79 (88) |
| Linseed oil | 22 | 22 | 22 | 22 | 22 | 22 | 22 |
| Palm kernel oil | 44 (48) | 44 (48) | 44 (48) | 44 (48) | 44 (48) | 44 (48) | 44 (48) |
| Nutrient premix* | 0 | 0.25 | 0.5 | 1.0 | 1.5 | 2.0 | 4.0 |
| Histidine | 0.00 | 0.34 | 0.68 | 1.36 | 2.04 | 2.72 | 5.44 |
| Cholesterol | 0.00 | 0.28 | 0.56 | 1.13 | 1.69 | 2.25 | 4.50 |

**Note:**
\* Times requirement. All diets were added 38 g kg$^{-1}$ monosodium phosphate, mineral additions were adjusted to each micronutrient premix (E.-J. Lock, 2016, unpublished data). All diets were added 9.3 g kg$^{-1}$ lysine, 1.8 g kg$^{-1}$ threonine, 8 g kg$^{-1}$ choline (50%), 0.25 g kg$^{-1}$ barox becp dry, 0.5 g kg$^{-1}$ yttrium oxide.

rest plant ingredients, Table 1). A NP was added in graded amounts, replacing some of the field peas in the formulation. Phosphate, lysine, threonine and choline were added to all diets in equal amounts. An antioxidant mixture to protect the feed during production, and yttrium for later digestibility measurements were also added to all diets. Diet acronyms were as follows: 0NP had no addition of the micronutrient premix, then the NP was added in graded amounts to the six diets called 25, 50, 100, 150, 200 and 400NP. The general idea was that the 100NP diets should contain 100% of the assumed requirement, achieved by addition of the NP to the basal diet (based on available data, primarily for rainbow trout (NRC, 2011) for each nutrient). However, some nutrients such as α-tocopherol and Se were already present at or above expected minimum requirements in the basal diet, but these were added in the NP to achieve increasing levels in the seven diets and to examine if higher requirements could be assessed. The 25NP diet contained 0.25 times the NP from the 100NP diet, while the 400NP contained four times the NP from the 100NP diet. As all nutrients were also present in the diet ingredients to some extent (not only contributed by the NP), the 400NP diet would never have four times higher content than the 100NP diet, and the fold difference would vary depending on the contribution from the NP versus the diet ingredients for each nutrient. The NP contained vitamin D$_3$, α-tocopheryl acetate, vitamin K$_3$, vitamin A$_1$, ascorbyl monophosphate, vitamin B$_6$, biotin, cobalamin, folate, pantothenic acid, riboflavin, thiamine, niacin, Se (as inorganic sodium selenite), iodine, copper, cobalt, manganese, zinc, crystalline DL-methionine, and taurine. Crystalline L-histidine and cholesterol were also added in graded amounts. The analyzed

**Table 2 Analyzed feed composition.** All results are the mean of two analytical parallels. Protein, lipid, starch ash and dry matter are given in g kg$^{-1}$ dry weight, energy as MJ kg$^{-1}$, while all other diet components are given as mg kg$^{-1}$. Slight difference in macronutrient composition; Trial 2 is given in parenthesis.

| | 0NP | 25NP | 50NP | 100NP | 150NP | 200NP | 400NP |
|---|---|---|---|---|---|---|---|
| *Proximate composition, g kg$^{-1}$ dw* | | | | | | | |
| Protein | 453 (480) | 469 (472) | 449 (440) | 456 (480) | 462 (480) | 470 (480) | 461 (480) |
| Lipid | 213 (220) | 203 (220) | 219 (210) | 211 (230) | 208 (220) | 197 (240) | 195 (220) |
| Starch | 112 | 112 | 109 | 104 | 106 | 107 | 94 |
| Ash | 66 | 68 | 66 | 67 | 69 | 60 | 75 |
| Dry matter | 910 (950) | 930 (940) | 920 (930) | 920 (930) | 930 | 920 (930) | 920 |
| Energy, MJ kg$^{-1}$ | 22.8 | 22.7 | 22.6 | 22.7 | 22.4 | 22.5 | 22.0 |
| *Micronutrients involved in redox regulation mg kg$^{-1}$ dw* | | | | | | | |
| *Trial1:* | | | | | | | |
| α-TOH | 48 | 72 | 84 | 118 | 153 | 193 | 339 |
| γ-TOH | 49 | 54 | 60 | 61 | 51 | 52 | 51 |
| Ascorbic acid | 4.7[1] | 21 | 43 | 86 | 143 | 192 | 351 |
| Selenium | 0.42 | 0.45 | 0.52 | 0.62 | 0.80 | 1.04 | 1.39 |
| *Trial2:* | | | | | | | |
| α-TOH | 76 | 91 | 109 | 141 | 178 | 230 | 239 |
| γ-TOH | 54 | 51 | 51 | 52 | 54 | 61 | 55 |
| Ascorbic acid | < 5.5 | 14 | 28 | 63 | 89 | 140 | 170 |
| Selenium | 0.47 | 0.48 | 0.56 | 0.79 | 0.91 | 1.0 | 1.1 |

**Note:**
[1] Below the limit of quantification, uncertain value.

composition of the diets can be found in Table 2, including proximate composition and contents of the micronutrients focused in this presentation.

## Feeding trials

Both feeding trials were conducted in accordance with Norwegian laws and regulations concerning experiments with live animals, which are overseen by the Norwegian Food Safety Authority.

Trial 1: The trial with parr in freshwater took place at the Institute of Marine Research (Matredal, 61°N, Western Norway). The salmon were hatched in February, and in June the salmon parr were randomly distributed in fifteen 400 litre (1 × 1 × 0.4 m$^3$) experimental tanks and acclimated for 1 week while being fed a commercial feed (Ewos). The trial commenced on July 3rd, with duplicate tanks for each diet with the exception of NP100 that was run in triplicates. Each tank contained 100 fish with mean initial body weight of 18.3 ± 2.2 g. The fish were fed *ad libitum* with continuous feeding from automated feeders during night and day. However, care was taken to limit overfeeding, due to uncertainties in the collection of uneaten feed at such a small pellet size. Collection and weighing of uneaten feed was conducted daily at 13:00, with the exception of weekends. The fish were exposed to continuous light, and oxygen saturation was

monitored on a regular basis and was never below 75%. The fish were reared in freshwater, but with seawater added as a buffer, creating a salinity of 1.1–1.3 g L$^{-1}$. The temperature was kept constant throughout the experiment, at 12.4 °C (SD ± 0.7). The total duration of the feeding trial was 12 weeks, e.g. 84 days.

Trial 2: Post-smolt Atlantic salmon were randomly distributed among fifteen sea cages (5 × 5 × 5 m; 125 m$^3$; 150 fish per cage) at Gildeskål Research Station, GIFAS, Gildeskål kommune, Norway (67°N, Northern Norway). Prior to the start of the trial, fish were acclimated to the environmental conditions for two weeks, the feeding trial started in January 2013. At start, the average fish weight was 228 ± 5 g and during the 157 day feeding period the fish more than doubled in weight. As in standard aquaculture practice, fish were reared under 24 h light regime before the start of the trial and during the first 3 months of the experiment. Cages were illuminated by four 400 W IDEMA underwater lights that were positioned at the center of each block of four cages at a depth of three meters. Fish were hand-fed to satiation two times daily and feed intake was recorded for each sea cage. Total feed intake and mortality were recorded daily. Water temperature, salinity and oxygen saturation over the course of the trial varied from 4.1 (January) to 10 °C (June), 30–34.2 g L$^{-1}$, and 8.7–12.2 mg/L, respectively.

## Sampling

Fish were anesthetized (Benzoak® VET, 0.2 ml/L, ACD Pharmaceuticals, Leknes, Norway) and killed by a blow to the head. In Trial 1, the fish in each tank were bulk weighed for total biomass at each sampling point and body weight and forklength were measured on individual fish, 5–44 fish depending on sampling point (sexually maturing males were excluded). In Trial 2, individual fish were weighed at each sampling (mid sampling 42 fish per cage, end sampling 34 fish per cage). In both trials, blood was drawn from the caudal vein (*Vena caudalis*) by means of a heparinized medical syringe from eight fish per tank or cage, before organs were dissected and kept as individual samples. Pooled organ samples based on 10 fish for each tank or cage (liver, gills, muscle) were frozen on dry ice and later homogenized, while individual organ samples were flash frozen in liquid nitrogen.

## Chemical analysis of diets, whole fish and organs

Multi-element determination in the feed and tissue samples was done by ICP-MS (inductively coupled plasma mass spectrometry) (*Julshamn et al., 1999*). High performance liquid chromatography (HPLC) was used for determination of Asc (*Mæland & Waagbø, 1998*) and tocopherols analysed according to (*CEN, 1999*). Thiobarbituric acid reactive substances (TBARS) was analyzed according to *Hamre et al. (2001)*. For the analysis of total (tGSH) and oxidised (GSSG) glutathione, supernatants were prepared from samples using a commercial kit (Prod. No. GT40, Oxford Biomedical Research, Oxford, UK) before analysed at 405 nm in a microplate reader (iEMS Reader Ms, Labsystems, Finland) as described by *Hamre et al. (2014)*.

## Gene expression analysis

Total RNA was purified from frozen liver using the EZ1 RNA Universal Tissue Kit on the BioRobot® EZ1 (Qiagen, Hilden, Germany), including the optional DNase treatment step in the protocol. Homogenisation in QIAzol lysis reagent from the kit was performed on the bead grinder homogeniser Precellys 24 (Bertin Technologies, Montigny-le-Bretonneux, France) for $3 \times 10$ s at 6,000 rpm. Quantity and quality of RNA were assessed with the NanoDrop® ND-1000 UV-Vis Spectrophotometer (NanoDrop Technologies, Wilmington, DE, USA) and a selection of samples were evaluated on the Agilent 2100 Bioanalyzer (Agilent Technologies, Palo Alto, CA, USA), with the 6,000 Nano LabChip® kit (Agilent Technologies, Palo Alto, CA, USA). Average RNA integrity number (RIN) for the samples in Trial 1 (39 samples) was $9.4 \pm 0.1$ (mean $\pm$ SEM, n = 24), in Trial 2, (90 samples): $9.5 \pm 01$ (mean $\pm$ SEM, n = 11).

Reverse transcription (RT) was performed on a GeneAmp Polymerase chain reaction (PCR) 9700 (Applied Biosystems (AB)) using the TaqMan® reverse transcriptase kit with oligo(dT) primers (AB). Primer sequences are given in Table 3. Samples were run in duplicate (500 ng, $\pm$ 5%), in addition to a six point dilution curve in triplicate (1,000–31.25 ng), non-template and non-amplification controls. Real-time PCR amplification and analysis were performed on a LightCycler 480 Real-time PCR system (Roche Applied Science, Basel, Switzerland) with SYBR® Green I Mastermix (Roche Applied Science). Pipetting of plates was done using a Biomek® 3000 Laboratory automation workstation (Beckman Coulter, Fullerton, USA). Thermal cycling was done for 45 cycles of 10 s at each of 95, 60 and 72 °C (basic programme from Roche), followed by melting analysis to confirm that only one product is present.

Cycle threshold (Ct)-values were calculated using the second maximum derivative method in the Lightcycler® software. Amplification efficiency was determined using the dilution curves with the formula E = 10 ^ (−1/slope), with the slope of the linear curve of Ct-values plotted against the log-dilution (*Higuchi et al., 1993*). Normalization to the reference genes and data analysis was conducted with GenEx 4.3.5 (MultiD Analyses AB, Gothenburg, Sweden) or with the geNorm applet (*Vandesompele et al., 2002*).

## Calculations

Nutrient retention = ((final biomass $\times$ final nutrient concentration) − (initial biomass $\times$ initial nutrient concentration)) $\times$ 100/(Total feed intake $\times$ nutrient concentration in feed).

The two-electron half-cell reduction potential of the 2GSH/GSSG redox-couple was calculated according to the Nernst equation:

$$E_h = E^{0\prime} - RT/nF \ln\left([GSH]^2/[GSSG]\right)$$

where the GSH and GSSG concentrations are in M and $E_h$ is given in volts. $E^{0\prime}$ is the standard reduction potential at pH7 and 25 °C and was assumed to be −0.240 V (*Kemp, Go & Jones, 2008*; *Schafer & Buettner, 2001*). The measurements are the average of whole organs and do not take into account that the reduction potential varies between cell

**Table 3  PCR assays.**

| Gene | Full gene name | Accession no. | Forward primer | Reverse primer | Amplicon size (bp) | PCR efficiency[a/b]* |
|------|----------------|---------------|----------------|----------------|--------------------|----------------------|
| cuznsod | CuZn superoxide dismutase | BG936553 | CCACGTCCATGCCTTTGG | TCAGCTGCTGCAGTCACGTT | 141 | 1.94/1.95 |
| mnsod | Mn superoxide dismutase | DY718412 | GTTTCTCTCCAGCCTGCTCTAAG | CCGCTCTCCTTGTCGAAGC | 227 | 1.96/2.06 |
| cat | Catalase | BG935638 | CCCAAGTCTTCATCCAGAAACG | CGTIGGGCTCAGTGTTGTTGA | 101 | 1.90/1.97 |
| gpx1 | Glutathione peroxidase 1 | DW566563 | GCCCACCCCTTGTTTGTGTA | AGACAGGGCTCCACATGATGA | 103 | 1.85/2.05 |
| gpx3 | Glutathione peroxidase 3 | DW561212 | TTCCCTCCAATCAGTTTGG | ATCCCCCCTCTGGAATAGCA | 123 | 1.97/2.05 |
| gr | Glutathione reductase | BG934480 | CCAGTGATGGCTTTTTTGAACTT | CCGGCCCCCACTATGAC | 61 | 2.03/1.97 |
| g6pd | Glucose-6-Phosphate dehydrogenase | > Contig7869_Atlantic_salmon | CTTTGGGCCAATCTGGAACA | TCCCGGATGATTCCAAAGTC | 114 | 2.08/2.00 |
| nfe2l2 | Nuclear factor (erythroid-derived 2)-like 2 | BT044699 | TCGCTGAAGGAGGAGGAAGGA | GTCCCTCAGCAGACGGAAAA | 120 | 1.83/2.05 |
| gclc | Glutamate-Cysteine ligase, catalytic subunit | > Contig16361_Atlantic_salmon | CGTCCTGCTCACCAGGGTTA | GCCCTCTTGGACTGCATTTC | 112 | 1.93/2.00 |
| actb | Beta-actin | BG933897 | CCAAAGCCAACAGGGAGAA | AGGGACAACACTGCCTGGAT | 92 | 1.93/20.00 |
| ee1ab | Eukaryotic translation elongation factor 1 alpha B | BG933853 | TGCCCCTCCAGGATGTCTAC | CACGGGCCCACAGGTACTG | 57 | 2.04/2.03 |
| uba52 | Ubiquitin A-52 Residue ribosomal protein fusion product 1 | NM_001141291 | CCAATGTACAGCGCGCCTGAAA | CGTGGCCATCTTGAGTTCCT | 110 | 2.04/- |

**Note:**

* a = Trial 1, b = Trial 2.

types, and between organelles within the cells (*Kemp, Go & Jones, 2008*; *Morgan et al., 2013*; *Schafer & Buettner, 2001*).

## Statistical analyses

GraphPad Prism (ver. 6.05) was used for the regression analyses. For the individual nutrients, the x-axis represented analyzed dietary concentrations, while for GSH and gene expression data the x-axis represented added NP. Since the nutrient concentrations in the 400% NP diet were outliers in Trial 2, the value of this diet was recalculated to 239% NP, the mean of the concentrations of α-TOH, Asc and Se. The best fit of the data after comparing different functions moving from simple to more complex relationships, was used. In figures with first order polynomials, p indicates the probability that the slope is equal to zero. In figures with second order fits, p gives the probability that the first order equation fits the data better than the second order equation. The logarithmic equation was applied for vitamin C retention in Trial 2 and p indicates the probability that a second order polynomial has a better fit than the logarithmic function. A function for overall comparison of slopes and intercepts of datasets fitted to linear equations, was used for comparison of data between Trials 1 and 2. The software also has a function for detecting outliers which was used for the data on muscle GSH/GSSG in post-smolts and on the gene expression data.

Correlation analyses of gene expression was performed using the software Statistica (ver. 11, Statsoft Inc., Tulsa, OK, USA) and Principle component analyses (PCA) plots were produced in Sirius (ver. 8.1, Pattern Recognition Systems AS, Bergen, Norway).

## RESULTS

Fish performance is reported in detail by *Hemre et al. (2016)*. Briefly, fish in Trial 1 grew from an initial weight of 18.3 g (± 2.2) to a range of 78.6 g (± 1.9) to 87.3 g (± 4.5). Both fish growth and protein retention increased with increasing dietary NP, while lipid retention decreased, together with liver index and viscera-somatic index. In Trial 2, initial fish size was 228 g (± 4.2) and average final weight was 482 g (± 17). There was no effect of the NP on growth or protein and lipid retention in this trial. Survival was high in both trials, close to 100%, and with no difference between diet groups.

The dietary concentrations of α-TOH, Asc and Se (Fig. S1) were well fitted to first order polynomials in Trial 1 ($R^2 > 0.98$). In Trial 2, the diet designated 400% was an outlier, while the other diets showed a good fit to a linear equation ($R^2 > 0.97$). The diet planned to contain 400% NP in Trial 2 was recalculated to contain 239% NP, based on the nutrient analyses of the diets. γ-TOH level was similar in all diets, as it was only derived from the feed ingredients (Fig. S1). The dietary concentrations at 100% NP were 118 and 141 mg kg$^{-1}$ for α-TOH, 67 and 63 mg kg$^{-1}$ for Asc and 0.62 and 0.79 mg kg$^{-1}$ for Se, in Trials 1 and 2, respectively. At 0% NP the dietary concentrations were 48 and 76 mg kg$^{-1}$ for α-TOH, 4.7 and < 5.5 mg kg$^{-1}$ for Asc and 0.42 and 0.47 mg kg$^{-1}$ for Se (Table 2).

There was a linear relationship between dietary and whole body concentrations of α-TOH in Trial 1 ($R^2 = 0.94$; p (slope = 0) $< 10^{-4}$; Fig. 1). In Trial 2, a second order polynomial fitted the data better than a first order polynomial (p = 0.03). The range of

## Trial 1

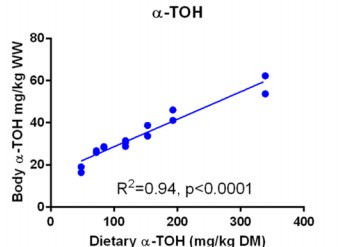
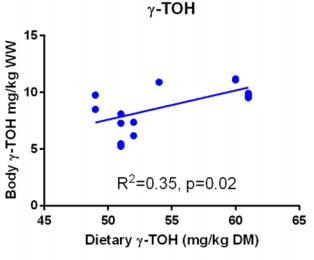
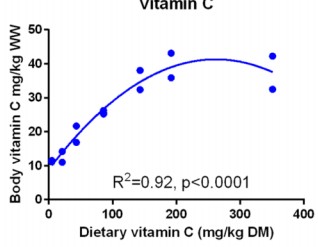
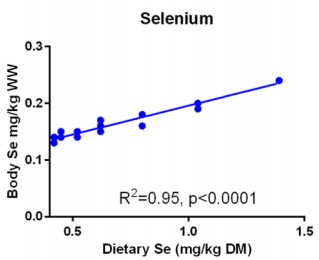

## Trial 2

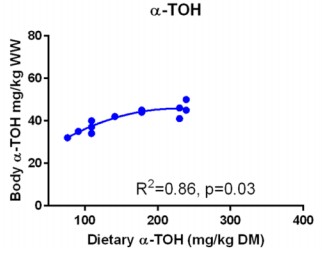
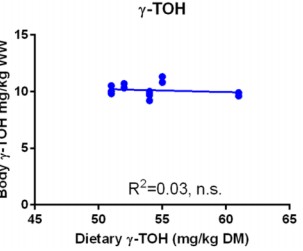
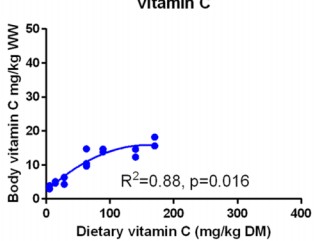
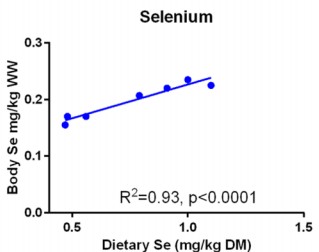

**Figure 1 Whole body concentrations (mg kg$^{-1}$ wet weight) of redox dependent micronutrients in Atlantic salmon parr (Trial 1) and post-smolt (Trial 2) in response to dietary supplementation of micronutrients and selected amino acids at supplementation of 0–400% NP.** Data were first fitted to second order polynomials and the fit was then compared to a first order polynomial fit, which was chosen when p > 0.05 for the second order equation. The p given in figures with first order and second order fits indicate that the slope is significantly different from 0 or that second order fits are significantly better than first order fits, respectively. n.s., not significant.

whole body concentrations was between 20 and 60 mg kg$^{-1}$ in Trial 1 and between 30 and 50 mg kg$^{-1}$ in Trial 2. Whole body $\gamma$-TOH concentration varied between 5 and 10 mg kg$^{-1}$, with a significant positive slope in Trial 1 (p = 0.02) and no effect of diet in Trial 2 (Fig. 1). Whole body Asc concentration followed a second order polynomial in both trials (Trial 1, $R^2$ = 0.92, p < 0.0001; Trial 2, $R^2$ = 0.88, p = 0.016). Whole body concentration plateaued at a dietary concentration of Asc of 190 mg kg$^{-1}$ DM and a whole body concentration of 39.5 mg kg$^{-1}$ in Trial 1. In Trial 2, the plateau was reached at dietary and whole body concentrations of 100 and 14 mg kg$^{-1}$, respectively. Whole body Se showed a linear relationship with the dietary concentration in both trials ($R^2$ > 0.93, p (slope = 0) < 10$^{-4}$). The diet dependent whole body concentration was significantly higher in Trial 1 than in Trial 2 (p < 10$^{-4}$).

The retention of $\alpha$-TOH was not affected by diet in Trial 1 and followed a second order polynomial in Trial 2 (p = 0.02) with the highest retentions at intermediate dietary $\alpha$-TOH concentrations (Fig. 2). Diet did not affect the retention of $\gamma$-TOH in Trial 1, while increasing the NP in Trial 2 had a negative effect on $\gamma$-TOH retention (p < 0.004). The range of retention of both $\alpha$- and $\gamma$-TOH was 20–30%. In Trial 1, retention of Asc ranged from 11–42% and was negatively linearly related to dietary Asc ($R^2$ = 0.34, p (slope = 0) = 0.02). In Trial 2, vitamin C retention was negative at dietary levels

## Trial 1

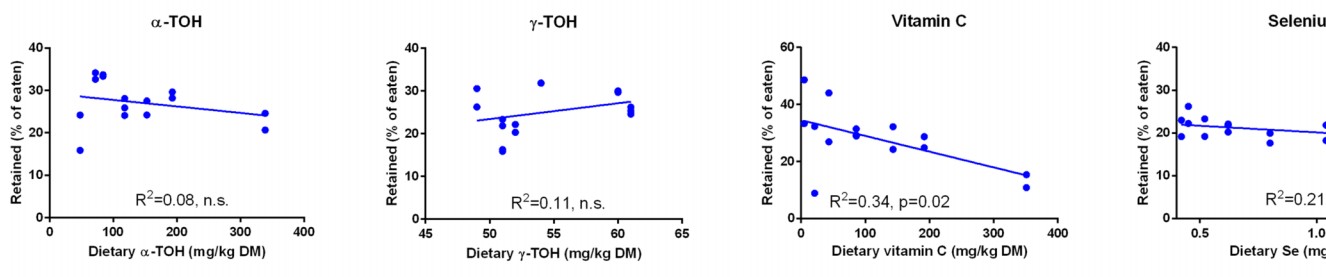

## Trial 2

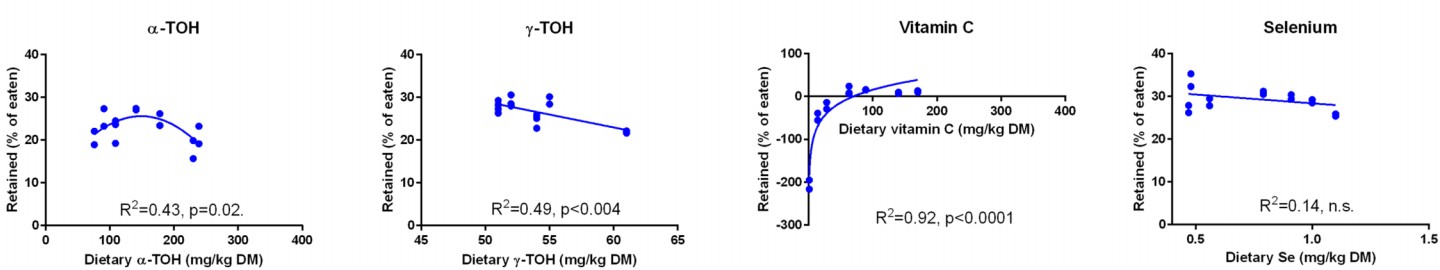

**Figure 2** **Retention (% of eaten) of redox dependent micronutrients in Atlantic salmon parr (Trial 1) and smolt (Trial 2) in response to dietary supplementation of micronutrients and selected amino acids at 0–400% NP.** Data were first fitted to second order polynomials and the fit was then compared to a first order polynomial equation fit. α-Tocopherol in Trial 2 had significantly best fit to the second order equation ($p < 0.05$). Vitamin C retention fitted a logarithmic equation better than a second order polynomial ($p < 0.001$). The p given in figures with first order fits indicates that the slope is significantly different from 0. n.s., not significant.

below 63 mg kg$^{-1}$ and max retention was 16% at 89 mg kg$^{-1}$. A logarithmic function was found to have the best fit to the data ($R^2 = 0.92$). Diet did not affect retention of Se, which ranged from 20 to 30%. The retention of Se was higher in Trial 2 than in Trial 1 ($p < 10^{-4}$).

GSH and GSSG concentrations were measured in muscle (Fig. 3) and liver (Fig. 4). In muscle in Trial 1, GSH concentrations ranged from 100 to 200 μmol kg$^{-1}$, GSSG ranged from 0 to 0.4 μmol kg$^{-1}$ with no effect of diet. There was a tendency that the resulting GSH based redox potential followed a second order polynomial at dietary NP below 150%, so that the tissue became transiently more reduced at supplementation of 25–100% NP than at 150–400% NP ($p = 0.058$). The redox potential of muscle in Trial 1 ranged from −240 to −190 mV. In Trial 2, muscle GSH and GSSG concentrations ranged from 43 to 80 and 0 to 0.4 μmol kg$^{-1}$, respectively, and the redox potential from −170 to −220 mV, with no effect of diet. The muscle GSH/GSSG concentrations in these fish were close to quantification limits and some of the data points were removed as outliers. Muscle GSH concentration was higher ($p < 0.0001$), the redox potential lower ($p = 0.002$) and GSSG concentration similar in Trial 1 compared to Trial 2. In the liver, GSH, GSSG and the GSH based redox potential were not affected by the diet, however there was a tendency towards a significant second order fit to the data on liver redox potential in Trial 1 ($p = 0.066$), implicating a higher redox potential in fish fed diets with intermediate NP

## Trial 1

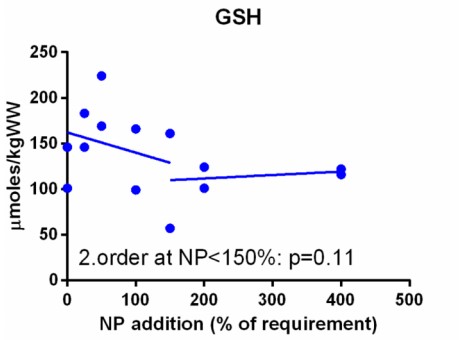
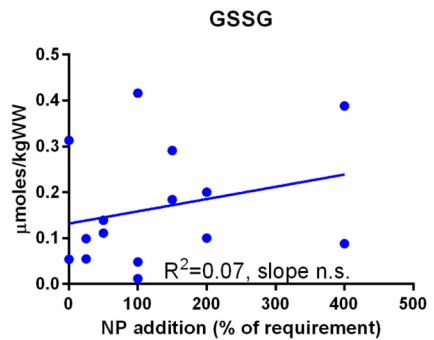
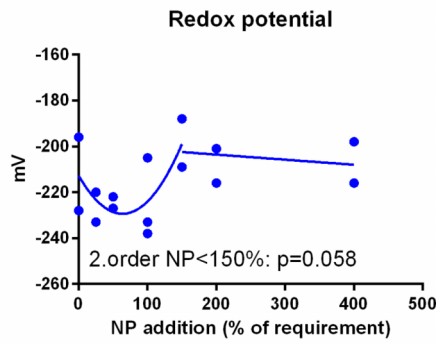

## Trial 2

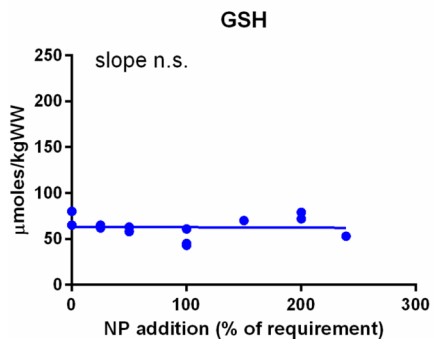
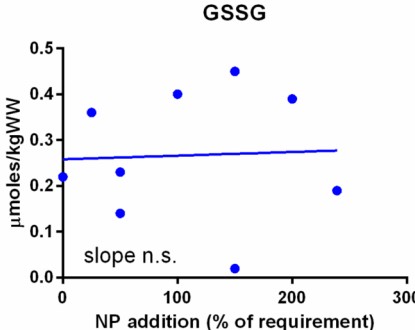
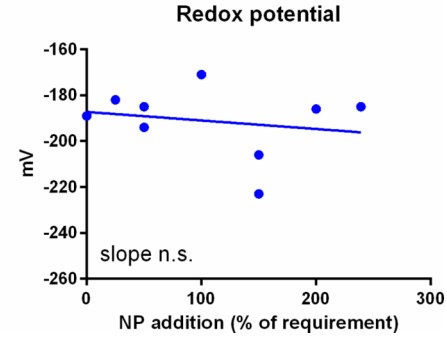

**Figure 3 Muscle reduced and oxidized glutathione (GSH and GSSG, μmoles kg$^{-1}$ wet weight) and the GSH based redox potential (mV) in Atlantic salmon parr (Trial 1) and post-smolt (Trial 2).** The dietary supplementation of micronutrients and selected amino acids was graded from 0–400% NP, where 100NP is the assumed requirement.

additions. Liver GSH concentration was higher ($p < 0.0001$), while GSSG concentration ($p = 0.01$) and the redox potential ($p < 0.0001$) were lower in Trial 1 than in Trial 2.

Muscle TBARS (Fig. 5) was negatively correlated to NP addition in both trials ($p = 0.004$ in Trial 1, $p = 0.03$ in Trial 2) and the absolute level was similar in the two trials ($p = 0.09$).

Expression of redox dependent genes was measured in the liver of individual salmon in both trials. PCA plots showed that the dietary groups were not well separated according to expression of the redox related genes (Fig. 6). However, in Trial 1 there was a tendency to grouping of samples related to dietary supplementation of NP, with NP supplementation correlating negatively to *g6pd*, *gclc*, *gpx1* and *gr* ($-0.38 < R^2 < -0.43$) and positively to *gpx3* ($R^2 = 0.39$). *Gr* showed the most covariation with other genes and correlated to *g6pd*, *gpx1* and *gclc* at $0.79 > R^2 > 0.49$ and to *mnsod* and *cuznsod* at $R^2 = 0.39$ and $0.34$, respectively. *Gr* was also correlated to *nfe2l2* ($R^2 = 0.52$), *cat* was not correlated to the other genes while *gpx3* was negatively correlated to the other genes, significant for *g6pd* ($R^2 = -0.33$). In Trial 2, diet was correlated to

## Trial 1

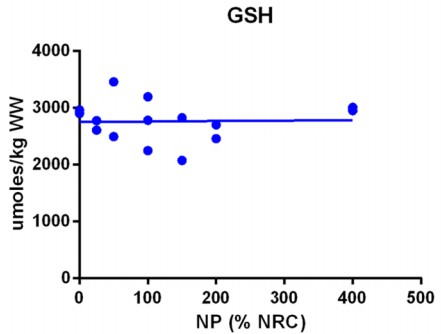
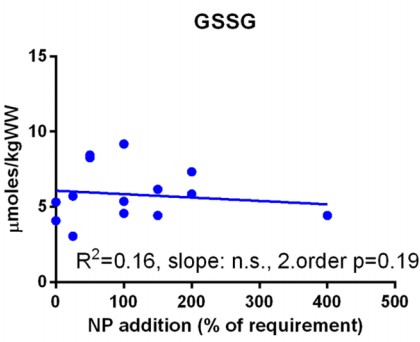
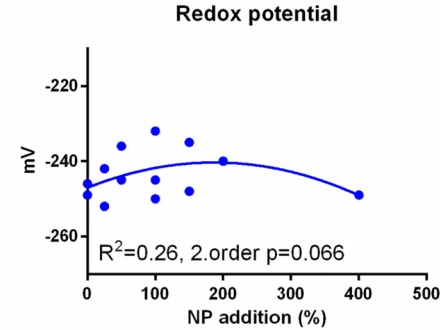

## Trial 2

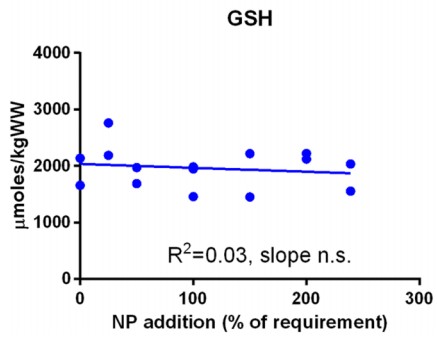
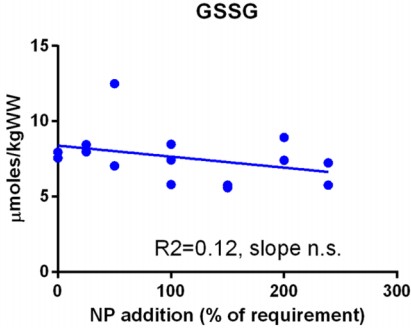
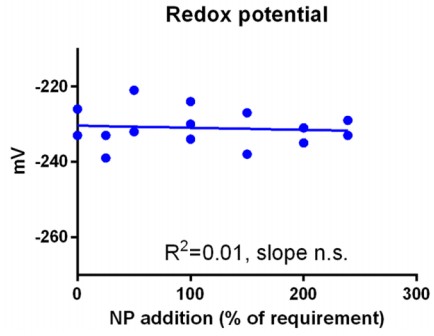

**Figure 4 Liver reduced and oxidized glutathione (GSH and GSSG, $\mu$moles kg$^{-1}$ wet weight) and the GSH based redox potential (mV) in Atlantic salmon parr (Trial 1) and post-smolt (Trial 2).** The dietary supplementation of micronutrients and selected amino acids was graded from 0–400% NP, where 100NP is the assumed requirement.

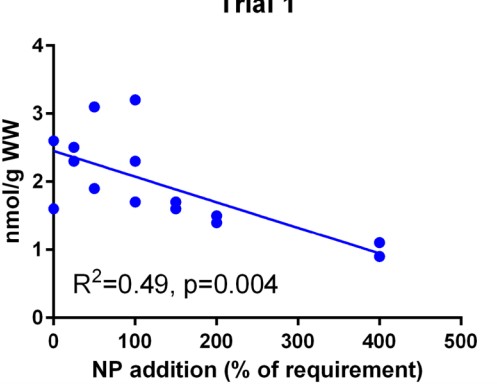
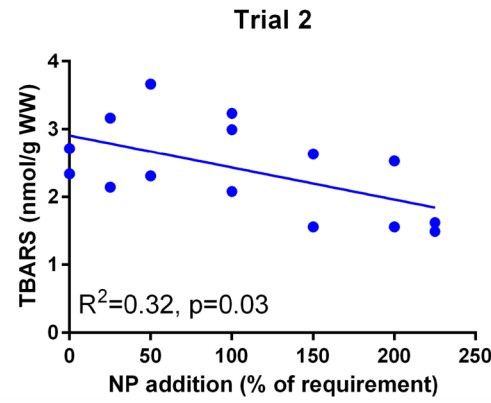

**Figure 5 Muscle TBARS (nmoles g$^{-1}$ ww) in Atlantic salmon parr (Trial 1) and post-smolt (Trial 2).** The dietary supplementation of micronutrients and selected amino acids was graded from 0–400% NP, where 100NP is the assumed requirement.

*cuznsod, mnsod* and *cat* (0.39 < R$^2$ < 0.24). The expression of *gr, g6pd, gpx1* and *gclc* comprised one distinct group and *mnsod, cat* and *cuznsod* another. *Gr* correlated with *g6pd, gpx1* and *gclc* at 0.67 > R$^2$ > 0.37, while *gr* correlation to *mnsod, cat*

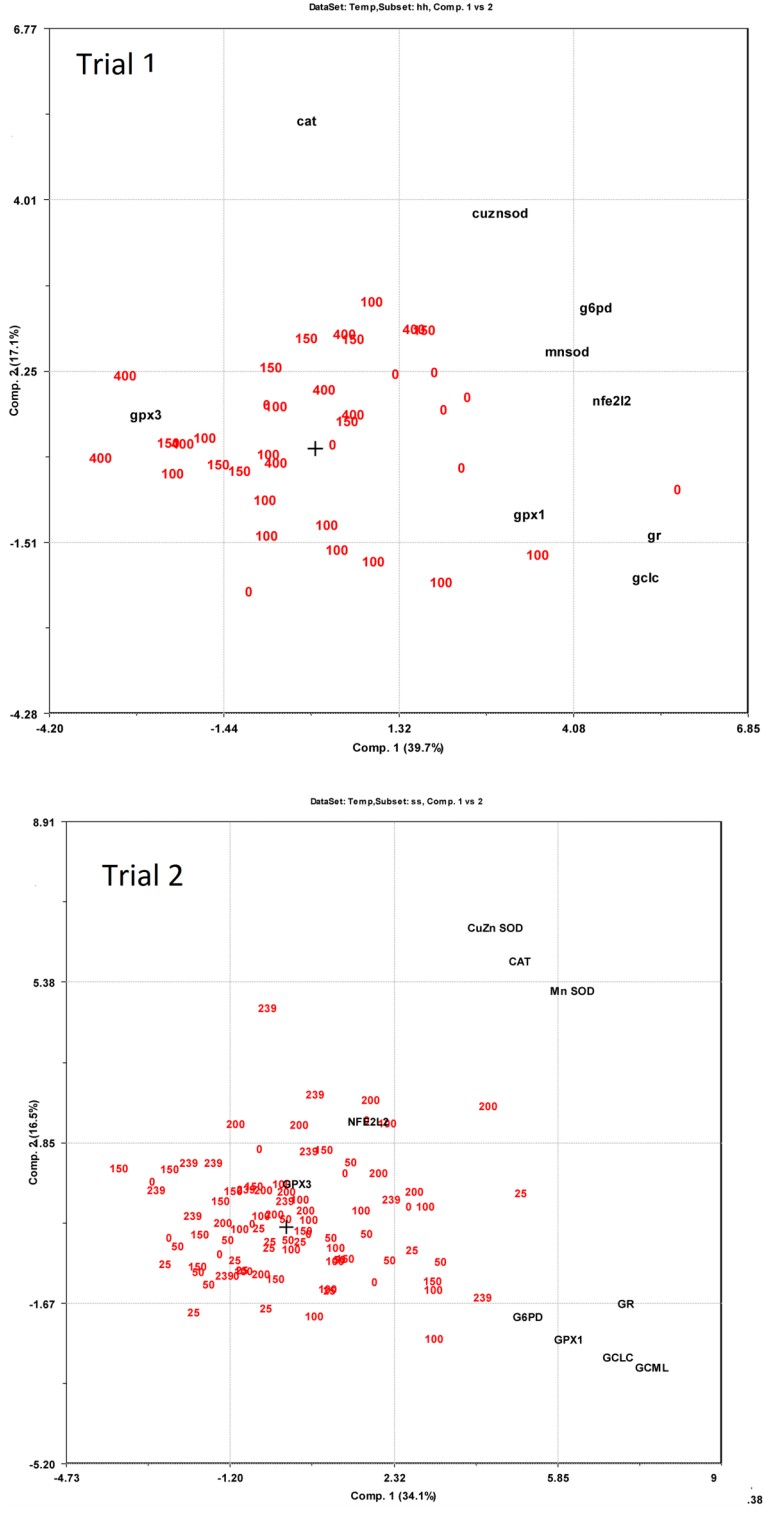

**Figure 6  PCA bi-plots on expression of redox dependent genes related to dietary supplementation of micronutrients and selected amino acids at 0–400% NP.**

and *cuznsod* was $0.32 > R^2 > 0.25$. *Gpx3* and *nfe2l2* were not significantly correlated to the other genes.

Relations between diet and gene expression are given in Fig. S1. In Trial 1, increasing supplementation of NP gave a reduction in *gclc*, *g6pd*, *gpx1*, *gr* and *mnsod* expression, while *nfe2l2*, *cat* and *cuznsod* expression were unaffected by diet. In Trial 2, data on *nfe2l2* and *gclc* were fitted to second order polynomials and had higher expression at intermediate NP supplementation. *Gpx1* showed the best fit when using a third order polynomial with peak expression at 50 and 100% NP inclusion. *Gpx3* had higher expression at low NP inclusion, while *mnsod* expression increased with increasing NP. *Cuznsod* expression was slightly lower at intermediate NP inclusion than at high and low inclusion, while *G6pd* and *gr* expression were unaffected by the diet. Average expression of *gpx1, gpx3* and *mnsod* was higher in Trial 2 than in Trial 1 ($p < 0.0002$). Increasing NP inclusion had opposite effects on *mnsod* expression in the two trials, which decreased in Trial 1 ($p = 0.04$) and increased in Trial 2 ($p = 0.002$).

## DISCUSSION

The present study showed differences in redox status between parr and post-smolts and effects of supplementation of the NP on redox regulation. However, there were no clear deficiencies of the antioxidant nutrients, vitamin C, vitamin E and Se, even in fish fed the unsupplemented diet. The reason may be that the plant-based feed ingredients contained relatively high levels of vitamin E and Se and that the experimental periods were too short for fish of these body sizes to develop vitamin C deficiency symptoms. The basal feed contained less than 5.5 mg kg$^{-1}$ Asc, which is below the minimum requirement in Atlantic salmon of 10–20 mg kg$^{-1}$; however, it took 18 weeks to deplete first-feeding Atlantic salmon for Asc (*Sandnes, Torrissen & Waagbø, 1992*) and a longer period of time may be needed to deplete larger fish sizes as the post-smolts. The final concentrations of α-TOH and Se in the basal feed were at or above minimum requirements at approximately 50 and 0.3 mg kg$^{-1}$ diet, respectively (*NRC, 2011*). Plant oils often have high concentration of tocopherols, however tocopherols are unstable during feed processing and storage and concentrations could vary considerably in the finished feeds (*Hamre, Kolås & Sandnes, 2010*; *Olsvik et al., 2011*). Se concentrations in plant ingredients vary according to Se concentration in soil (*Alfthan et al., 2015*). Therefore, even if α-TOH and Se may be sufficient in some plant ingredients, they should be added to fish feeds to ensure safe supplementation. Commercial fish feeds are commonly supplemented with higher levels of antioxidant nutrients than the minimum requirements given by *NRC (2011)*, which will protect the fish in periods with oxidative stress, such as high water temperatures (*Martínez-Álvarez, Morales & Sanz, 2005*) elevated water oxygen levels (*Lygren, Hamre & Waagbø, 2000*), vaccination (*Lygren, Hjeltnes & Waagbø, 2001*) and disease (*Trichet, 2010*; *Waagbø, 2006*). Accordingly, the diets with 100% NP used in this study, contained 118–141 mg kg$^{-1}$ α-TOH, 63–67 mg kg$^{-1}$ Asc and 0.62–0.79 mg kg$^{-1}$ Se, respectively.

The most sensitive biomarker of vitamin E deficiency in salmonids is lowered hemoglobin (Hb) concentration (*Hamre, 2011*). There was a slight positive correlation

between NP addition and Hb concentration in fish from Trial 1 (p (slope = 0) = 0.03) (*Hemre et al., 2016*), however, the slope was not significantly different from 0 after omitting the 400% NP group (p = 0.44), so fish fed the diets with low NP supplementation were clearly not vitamin E deficient. In Trial 2, there was no effect of diet on Hb (*Hemre et al., 2016*). *Hamre & Lie (1995)* found that mortality due to vitamin E deficiency commenced at whole body levels of α-TOH of 5 mg kg$^{-1}$ wet weight, while the whole body concentration in fish fed the basal diet in this study were 18 and 32 mg kg$^{-1}$ in Trials 1 and 2, respectively. Accordingly, none of the fish groups in this study experienced vitamin E deficiency, even though the low level of vitamin C in the diet without NP would make the fish more susceptible to low dietary vitamin E (*Hamre et al., 1997*).

The diets were fed for 12 and 22 weeks in Trials 1 and 2, respectively. The twelve weeks feeding period in Trial 1 should just be enough for the fish to adjust to the α-TOH concentrations of the diet according to previous reports (*Hamre, 2011*; *Hamre & Lie, 1995*). However, the slope of whole body concentration to dietary concentration was lower (0.13 vs 0.25) and the intercept with the y-axis was higher (15.6 vs 0) than previously reported, indicating that the steady state whole body α-TOH concentration had not yet been obtained (*Hamre, 1995*). A relation of body to dietary concentrations of α-TOH has until now only been measured in very young fish and these relationships were always linear (*Hamre et al., 1997*; *Hamre & Lie, 1995*). In post-smolts in the present study, the relationship followed a second order polynomial. It is possible that uptake of α-TOH is changing over development, moving from a strict linear relationship just after first-feeding to a saturable mechanism at later stages. If this is the case, the diet 100% NP with 141 mg kg$^{-1}$ α-TOH would be sufficient to saturate the system, while supplementation above this gave a decreased retention. In summary, supplementing commercial diets with above 150 mg kg$^{-1}$ α-TOH is recommended to promote fish health and welfare. A fish feed surveillance program run by NIFES in 2015 reported 185–320 mg kg$^{-1}$ α-TOH in 18 commercial salmon feeds, while two feeds contained as high as 570 mg kg$^{-1}$ (M. Sanden, 2016, unpublished data) A variable part of this α-TOH would have been derived from the feed ingredients. One have to take into account that α-TOH in the feed ingredients is in the free form, which is easily broken down in salmon intestine and therefore has quite low retention (*Meland, 1995*). It is therefore a risk involved in including ingredient contribution when deciding dietary α-TOH supplementation.

The non-α-TOHs cannot substitute for α-TOH, because of their lower biological activity (*Hamre, 2011*; *Kayden & Traber, 1993*), however they may have specific biological functions in cell signaling (*Golli & Azzi, 2010*; *Wallert et al., 2014*). In both trials of this study, the retentions of α- and γ-TOH were similar at 20–30%. Due to lower biological activity, retention of γ-TOH should be lower than α-TOH, but the result can be explained by similar retention of α- and γ-TOH in muscle and adipose tissue, which are the main constituents of the whole body sample. Retention of γ-TOH in other vital organs of salmon, such as gonads, intestine, and gills, is less than 30% compared to α-TOH (*Hamre & Lie, 1997*).

The whole body concentration of Asc followed a second order equation with respect to Asc supplementation in both trials. It is well known that tissue levels of Asc level off at dietary supplementation that is much higher than the levels where traditional deficiency symptoms are displayed (*Gabaudan & Verlhac, 2001*; *Waagbø & Sandnes, 1996*). Since Asc above this minimum requirement has positive effects on immune function, stress resistance and reproduction (*Gabaudan & Verlhac, 2001*; *Waagbø, 1994*; *Waagbø, 2006*), it may be fruitful to supplement more of the vitamin. The whole body concentration levelled off at a dietary Asc concentration of 190 mg kg$^{-1}$ in Trial 1 and at 63–89 mg kg$^{-1}$ in Trial 2. The maximum body concentration of Asc in fish from Trial 2 was approximately 30% of that in fish from Trial 1. While the data from Trial 1 follow the expected trajectory based on the literature, the fish in Trial 2 seems to have had an extraordinary high consumption of Asc, with negative retention for fish supplemented below 63 mg kg$^{-1}$ and maximum retention of 16% at higher supplementation, compared to max 42% retention in Trial 1.

There is probably an optimal ratio between supplementation of α-TOH and Asc, based on the hypothesis that vitamin E is recycled by vitamin C. A buildup of tocopheroxyl radicals would occur if too little vitamin C is present to recycle it, which would have a pro-oxidant effect on the tissues (*Hamre et al., 1997*). An indication of this principle is the higher mortality due to vitamin C deficiency in fish fed 300 compared to 150 mg kg$^{-1}$ vitamin E (*Hamre et al., 1997*). Another indication is that vitamin E and C in tissues from copepods and wild Ballan wrasse are 100 and 500 mg kg$^{-1}$ dry matter and 9 and 18 mg kg$^{-1}$ wet weight, respectively (*Hamre et al., 2013*; *Hamre et al., 2008*). Theoretically, one should supplement more Asc than vitamin E, although the exact optimal ratio has not yet been identified. Seventeen commercial salmon feeds analyzed in a fish feed surveillance program performed at NIFES in 2015, contained between 118 and 418 mg kg$^{-1}$ Asc, one feed contained 800 and two feeds contained more than 1,300 mg kg$^{-1}$ Asc. The concentration of Asc in the feeds was frequently below the concentration of α-TOH (M. Sanden, 2016, unpublished data).

Whole body Se concentration followed a first order relationship with the dietary concentration both in parr and post-smolt, and the retention was not affected by NP supplementation. A similar relationship was found in zebrafish, but this species had higher whole body Se concentration at similar supplementation levels. The difference may be species dependent or perhaps due to the use of Se enriched yeast containing mainly selenomethionine as the Se source in the zebrafish (*Penglase et al., 2014*) and selenite (Na$_2$SeO$_3$) here (*Lorentzen, Maage & Julshamn, 1994*). The basal diet contained 0.4–0.5 mg kg$^{-1}$ Se, while the requirement in rainbow trout is estimated at 0.15–0.38 mg kg$^{-1}$ (*Hilton, Hodson & Slinger, 1980*). It is therefore unlikely that fish fed the diets used in this study would become Se deficient. Judged by retention efficiency, the Se present in the feed ingredients had similar bioavailability to the added selenite. If *gpx1* expression had been affected by low Se in this study, increased NP would probably have increased this response (*Hilton, Hodson & Slinger, 1980*; *Penglase et al., 2014*). However, *gpx1* was not expressed according to this hypothesis in either trial. Retention and body concentrations of Se were higher in seawater than in fresh water, possibly caused by

differences in waterborne Se concentrations. Se toxicity was probably not encountered in the present study, since the diet with the highest NP inclusion contained 1.1 and 1.4 mg kg$^{-1}$ Se, which is below levels reported to be toxic in fish (*Penglase et al., 2014*).

To examine whether the graded dietary NP affected redox regulation in the salmon, we measured GSH/GSSG concentrations in liver and muscle and calculated the resulting $E_{GSH}$. In Trial 1, there was a tendency (p = 0.06–0.07) that NP supplementation below 150% gave more reduced fish, resulting from higher concentrations of GSH. Considering the above discussion, it is unlikely that this effect would have been caused by low levels of vitamin C, E or Se. The diets also had graded levels of taurine and methionine (*Hemre et al., 2016*) and at low levels of these nutrients, GSH synthesis or recycling and a resulting lowered $E_{GSH}$ could have been stimulated to protect against oxidation induced by low taurine (*Jong, Azuma & Schaffer, 2012*; *Penglase et al., 2015*), consistent with the relatively higher liver *gclc*, *g6pd* and *gr* expressions at low NP inclusion. In Trial 2, there were no effects of diet on GSH/GSSG concentrations and no correlation between diet and expression of the genes coding for GSH metabolizing enzymes. The higher muscle TBARS at low NP inclusion in both trials confirm that fish fed low levels of micronutrients and selected amino acids may have become oxidized. Overall, the results confirm that the tissue GSH/GSSG concentrations are relatively stable within experiments with salmon, even with large variations in dietary composition, as found by *Hamre et al. (2010)*.

However, the GSH/GSSG concentrations and $E_{GSH}$ were different in Trials 1 and 2, the fish in Trial 2 being more oxidized, with less GSH both in liver and muscle and more GSSG in the liver, than the fish in Trial 1. Expression of liver *gpx1* (cytosol and mitochondria) and *gpx3* (extracellular) was also higher in Trial 2, indicating excess activity in removal of $H_2O_2$ at the expense of GSH. This corresponds with an increased consumption of Asc in Trial 2. These results indicate that the fish in Trial 2 experienced some sort of oxidative stress. The fish did not seem to be diseased, since the growth and feed intake were good and mortalities were negligible (*Hemre et al., 2016*). The feed was not analyzed for oxidation, however, oxidized feed most often leads to reduced vitamin E retention (*Baker & Davies, 1997*; *Hung, Cho & Slinger, 1981*), which was not encountered in the present study. The samples in Trial 2 were taken in June, with rapidly increasing water temperatures and longer day light, resulting in strong growth stimulation in salmon. In cod larvae fed copepods, a similar growth stimulation correlated with more oxidized fish having a higher whole body $E_{GSH}$ (*Karlsen et al., 2015*; *Penglase et al., 2015*). Based on these observations, we hypothesize that redox regulation is involved in surges in growth.

## CONCLUSIONS

This study gives no indications that diets rich in plant ingredients increase the vitamin C, E or Se requirements in Atlantic salmon. Feeds produced using the current feed ingredients, seem to have sufficient amounts of α-TOH and Se from the feed ingredients alone. However, due to possible variations in ingredient quality, feed processing and farming conditions, feeds should be supplemented with above 150 mg kg$^{-1}$ α-TOH equivalents. The Se in plant-based feed ingredients may vary and Se supplementation may

be warranted in some cases. However, the legal upper limit of Se in fish feeds in the European Union is 0.5 mg kg$^{-1}$ (Regulation (EC) No 1831/2003) and the law comes into force if the feeds are supplemented. Fish fed the highest concentration of Se (1.4 mg kg$^{-1}$) did not show any signs of toxicity. Based on body concentrations, Asc supplementation should be above 190 mg kg$^{-1}$, where the whole body Asc concentration leveled off in Trial 1. At this supplementation level, Asc deficiency can be avoided in periods with oxidative stress and optimal immune function and stress resistance can be obtained. Redox regulation, which includes differential consumption of antioxidant nutrients, seems to change during the production cycle of Atlantic salmon.

## ACKNOWLEDGEMENTS

We want to thank the technical staff at NIFES, IMR and GIFAS for great work with the practical aspects of this study.

### Funding

This study was part of the project "Advanced Research Initiatives for Nutrition & Aquaculture" (ARRAINA). Funded by EU, FP7 under the subprogramme "Food, Agriculture and Fisheries, Biotechnology" (FP7-288925). Additional funding was obtained from NIFES. The funders had no role in study design, data collection and analysis, decision to publish, or preparation of the manuscript.

### Grant Disclosures

The following grant information was disclosed by the authors:
Advanced Research Initiatives for Nutrition & Aquaculture" (ARRAINA).
EU, FP7 under the subprogramme "Food, Agriculture and Fisheries, Biotechnology": FP7-288925.
NIFES.

### Competing Interests

Kristin Hamre is an Academic Editor for PeerJ. Joana Silva is employed by Biomar AS, Trondheim, Norway. Bente Torstensen is employed by Marine Harvest ASA, Bergen Norway. Johan Johansen is employed by GIFAS AS, Indyr, Norway. Otherwise there are no competing interests.

### Author Contributions

- Kristin Hamre conceived and designed the experiments, analyzed the data, wrote the paper, prepared figures and/or tables, reviewed drafts of the paper.
- Nini H. Sissener conceived and designed the experiments, performed the experiments, wrote the paper, prepared figures and/or tables, reviewed drafts of the paper.
- Erik-Jan Lock conceived and designed the experiments, performed the experiments, wrote the paper, reviewed drafts of the paper.

- Pål A. Olsvik analyzed the data, contributed reagents/materials/analysis tools, wrote the paper, prepared figures and/or tables, reviewed drafts of the paper.
- Marit Espe conceived and designed the experiments, performed the experiments, wrote the paper, reviewed drafts of the paper.
- Bente E. Torstensen conceived and designed the experiments, wrote the paper, reviewed drafts of the paper.
- Joana Silva conceived and designed the experiments, performed the experiments, contributed reagents/materials/analysis tools, wrote the paper, reviewed drafts of the paper.
- Johan Johansen performed the experiments, contributed reagents/materials/analysis tools, wrote the paper, reviewed drafts of the paper.
- Rune Waagbø conceived and designed the experiments, performed the experiments, wrote the paper, reviewed drafts of the paper.
- Gro-Ingunn Hemre conceived and designed the experiments, performed the experiments, wrote the paper, reviewed drafts of the paper.

### Animal Ethics

The following information was supplied relating to ethical approvals (i.e., approving body and any reference numbers):

Both feeding trials were conducted in accordance with Norwegian laws and regulations concerning experiments with live animals, which are overseen by the Norwegian Food Safety Authority. Permission for these specific experiments were given by the Directorate of Fisheries, and accepted for feeding trials at GIFAS, §13 (Akvakulturloven) and §28a (Lakseforskriften) (ref 13/11363), and acknowledged by the advisory board 27.11.12 (ref: ARRAINA regression trial permission).

### Data Deposition

All raw data appear as points in the graphs.

### Supplemental Information

Supplemental information for this article can be found online at http://dx.doi.org/10.7717/peerj.2688#supplemental-information.

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
