# Peer review of "Antioxidant nutrition in Atlantic salmon (Salmo salar) parr and post-smolt, fed diets with high inclusion of plant ingredients and graded levels of micronutrients and selected amino acids"

_PeerJ, doi:10.7717/peerj.2688_

## Round 0.1 · original submission · Minor Revisions

Please address all changes as requested by the reviewers.

Reviewer 1 ·

Basic reporting

Well designed study and relevant work with a very balanced and integral use of expressions and good English grammar. It is well understood and detailed with good explanations of fundamental rationale leading to the study.

Experimental design

A sound and robust investigation that conforms to the science of fish nutrition. The use of fish, namely salmon and the feeding protocols and the diet formulations are most interesting.
The focus on micro-nutrients and anti-oxidant status of fish receiving high plant diets is adding to our knowledge base with novel approaches particularly the addition of genomics and molecular biology in appropriate areas to enhance the full suite of redox and other related oxidative stress measurements and vitamin and trace element requirements in a series of test feed formulations..

Validity of the findings

the finding are in keeping with some expectations but do enhance our comprehension of anti-oxidant micro- nutrients in the nutrition of farmed salmon. The work is scientifically robust and with sound statistical relevance to the study.
The results add weight to our growing knowledge base in aquaculture nutrition.

Additional comments

None

Reviewer 2 ·

Basic reporting

The article is well written and generally free of mistakes, The introduction is well planned and provides good background on each of vitamin E, Asc, Se and redox. It is a little long. The methods are well structured and logical. There are minor issues with explanations on feed collection (line 194) and measurement of fish (line 216) which can easily be corrected. Some description of tank versus cage do not seem consistent (line 219 and 220). There are minor issues with brackets and the citing of authors at the end of some sentences; e.g. line 228 and 231. Results were succinct and flowed easily from the methods. Discussion was informative, however still a little long. The reader is directed to many unpublished studies throughout the manuscript which is a somewhat frustrating.

Tables are OK. The authors should indicate whether formulations and chemistry is presented on as-is basis or dry-basis. Also check units for gross energy.

All figures (except for figure 5) were very small when printed for reading. If possible the size of figures should be increased (or less per page) to improve their interpretation by the reader.

References cited were extensive and relevant to the work.

Experimental design

The methods were adequately explained and provide enough detail to be reproduced. From above; the methods are well structured and logical. There are minor issues with explanations on feed collection (line 194) and measurement of fish (line 216) which can easily be corrected. Some description of tank versus cage (i.e. from trial 1 or trial 2) do not seem consistent (line 219 and 220).

Presenting some of the basic fish performance data in a table would be useful. The authors indicate this will be done elsewhere (Hemre et al. in-prep), but the mention of recording liver and gutted weights (line 222) leads the reader to expect these types of data will be presented in the current manuscript.

The statistical section seems a bit wordy and the authors might simplify their approach by stating the line of best fit was used, whether that be a linear or quadratic function. It would be useful to present data on the fitted coefficients, slopes and intercepts as well as confidence intervals around these values. Some reason for using the selected models would also be useful; i.e. what is the biological basis for fitting these types of curves to this type of data?

Validity of the findings

No comments

Additional comments

As indicated in sections above.

---

## Round 0.2 · Minor Revisions

Thank you for the revised document. I was hoping you would address the comments made by the reviewer, you seemed to dismiss most. Could you please make as many suggested changes as possible? Could you please confirm that you really checked all his queries and all this information is now correct?

I'm concerned about papers in press being cited, I think you either have to include all information here or wait for the other paper to be accepted.

---

## Round 0.3 · accepted · Accept

Once again thank you for submitting your manuscript to PeerJ and making all revisions in a timely manner.